# SparseDFF: Sparse-View Feature Distillation for One-Shot Dexterous Manipulation

**Qianxu Wang**[1,3]**, Haotong Zhang**[1]**, Congyue Deng**[2,✉]**, Yang You**[2]**,**
**Hao Dong**[1]**, Yixin Zhu**[3,4,✉]**, Leonidas Guibas**[2,✉]

[1] CFCS, School of Computer Science, Peking University, China

[2] Department of Computer Science, Stanford University, USA

[3] Institute for AI, Peking University, China

[4] PKU-WUHAN Institute for Artificial Intelligence, China

✉ congyue@stanford.edu, yixin.zhu@pku.edu.cn, guibas@stanford.edu

https://helloqxwang.github.io/SparseDFF

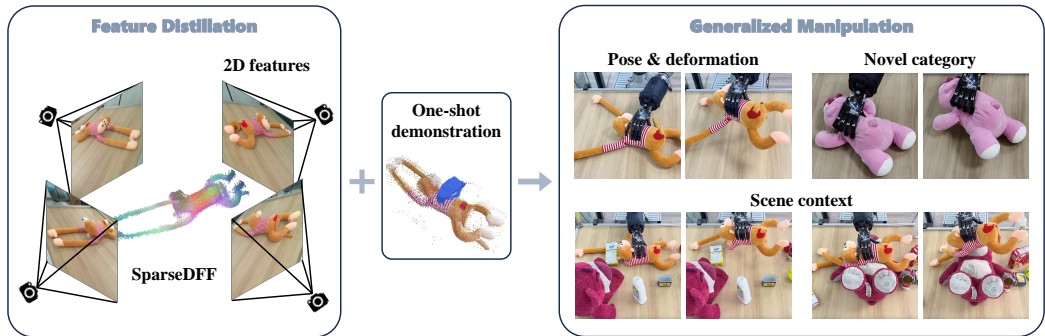

Figure 1: **Overview of SparseDFF.** We introduce a novel method, SparseDFF, for distilling **view-consistent** 3D Distilled Feature Field (DFF) from sparse RGBD images, readily **generalizable** to novel scenes without any modifications or fine-tuning. The DFFs create dense correspondences across scenes, enabling **one-shot** learning of dexterous manipulations. This approach facilitates seamless manipulation transfer to new scenes, effectively handling variations in object poses, deformations, scene contexts, and categories.

## Abstract

Humans demonstrate remarkable skill in transferring manipulation abilities across objects of varying shapes, poses, and appearances, a capability rooted in their understanding of semantic correspondences between different instances. To equip robots with a similar high-level comprehension, we present SparseDFF, a novel DFF for 3D scenes utilizing large 2D vision models to extract semantic features from sparse RGBD images, a domain where research is limited despite its relevance to many tasks with fixed-camera setups. SparseDFF generates **view-consistent** 3D DFFs, enabling efficient **one-shot** learning of dexterous manipulations by mapping image features to a 3D point cloud. Central to SparseDFF is a feature refinement network, optimized with a contrastive loss between views and a point-pruning mechanism for feature continuity. This facilitates the minimization of feature discrepancies w.r.t. end-effector parameters, bridging demonstrations and target manipulations. Validated in **real-world** scenarios with a dexterous hand, SparseDFF proves effective in manipulating both rigid and deformable objects, demonstrating significant **generalization** capabilities across object and scene variations.

## 1 Introduction

Learning from demonstration is a powerful approach for quickly imparting complex skills to robots. Although recent advancements have shown promising results in applying reinforcement learning to dexterous manipulation tasks (Xu et al., 2023; Wan et al., 2023; Li et al., 2023a), the effectiveness of these methods is often contingent on the availability of a carefully curated demonstration dataset and struggles to accommodate the varied demands of different tasks. Moreover, these techniques primarily target the manipulation of rigid objects and encounter significant obstacles in real-world applications and in scaling to datasets beyond those they were trained on. In stark contrast, humans demonstrate remarkable abilities to extrapolate and generalize from observed demonstrations (Lake

& Baroni, 2023; Jiang et al., 2023; Li et al., 2024b;a; 2023b; Xie et al., 2021; Lake et al., 2015). For instance, learning to hold a cat by watching someone do so can effortlessly extend to holding various other cats of different breeds, sizes, and appearances or even to entirely different animals, such as dogs, otters, or baby tigers, provided they are accessible and amenable. This exceptional capacity for generalization stems from the ability to discern underlying similarities across different instances, despite variations in appearance, pose, or species (Zhu et al., 2020; Fan et al., 2022).

To empower autonomous agents with human-like comprehension and generalization from demonstrations, leveraging object and scene representations from large vision models proves to be effective. Despite the prevalence of these models being trained on 2D imagery—attributed to the hurdles in acquiring and annotating 3D data—applying them directly to complex manipulation tasks, such as those requiring dexterous manipulation, remains a significant challenge. Recent endeavors have introduced DFFs (Kobayashi et al., 2022), which transform dense 3D feature fields from 2D image features, thus enhancing understanding of 3D scenes (Kerr et al., 2023) and facilitating interactions (Shen et al., 2023; Lin et al., 2023; Rashid et al., 2023; Ze et al., 2023).

Nevertheless, the predominant methodology for constructing these feature fields in 3D vision—often by leveraging techniques akin to NeRF—results in a heavy dependence on dense camera views for 2D-3D distillation (Shen et al., 2023; Rashid et al., 2023). This dependency constrains interaction scenarios to those with only sparse views available and impedes rapid training and inference, significantly narrowing the model's applicability. Additionally, the operations in existing works are relatively rudimentary, typically involving a gripper handling rigid objects, which points to the feature field's limited effectiveness (Simeonov et al., 2022; Shen et al., 2023; Lin et al., 2023; Rashid et al., 2023; Ze et al., 2023).

In this work, we present SparseDFF, a novel approach to generating view-consistent 3D DFFs from sparse RGBD observations, facilitating one-shot learning of dexterous manipulations transferable to new scenes. Our principal insight is that the main limitation for feature fields in manipulation is not a lack of visual information or the expressive capacity of the field model, but rather the consistency of local features. We show that a point cloud-based feature field, with enhancements in feature consistency, can provide precise optimization for a 24 DoFs dexterous hand.

More specifically, we project image features onto a 3D point cloud, facilitating their propagation across 3D space to form a dense feature field. At the heart of SparseDFF lies a lightweight feature refinement network trained only based on a single demonstration, optimized using a contrastive loss between pairwise views. Furthermore, we introduce a point-pruning strategy to improve feature continuity within each local area. The resultant feature fields create dense correspondences across varying scenes, enabling the establishment of an energy function on the end-effector pose from the original demonstration to the target manipulation. This approach allows for the one-shot learning of dexterous manipulations, adaptable to new scenes with variations in object poses, deformations, scene settings, or even differing object categories. Our methodology is validated through real-world experiments with a dexterous hand interacting with both rigid and deformable objects, exhibiting strong generalizations across various objects and scene settings.

To summarize, our contributions are threefold:
- We introduce a novel framework for **one-shot learning** of dexterous manipulations, leveraging semantic scene understanding distilled into 3D feature fields.
- We devise an efficient method for deriving **view-consistent 3D features** from 2D image models, incorporating a lightweight feature refinement network and a point pruning mechanism. This facilitates the application of our network to new scenes, predicting consistent features *without* requiring any adjustments or fine-tuning.
- Our **real-world** experiments with a dexterous hand affirm our method's effectiveness, demonstrating its robustness and superior **generalization** ability in varied scenarios.

## 2 RELATED WORK

**Implicit Fields for Manipulation** The identification of point-wise correspondences facilitates the transfer of manipulation policies across diverse objects. In contrast to methods based on key points (Manuelli et al., 2019; Xue et al., 2023; Florence et al., 2018), recent efforts focus on developing dense feature fields through implicit representations (Simeonov et al., 2022; Wu et al., 2023a; Ryu et al., 2022; Dai et al., 2023; Simeonov et al., 2023; Weng et al., 2023; Urain et al., 2023; Zhao et al.,

2022; Wu & Zhao, 2022). The integration of large vision models has propelled research towards leveraging DFFs for enabling few-shot or one-shot policy learning, 6-DOF grasps (Shen et al., 2023), sequential actions (Lin et al., 2023), and language-guided manipulations (Rashid et al., 2023). Yet, these approaches either depend on dense view inputs (Shen et al., 2023; Rashid et al., 2023), requiring extensive camera movement around the scene, or utilize single-view features (Lin et al., 2023), which may suffice for parallel grippers but fall short for dexterous manipulations due to spatial complexities.

Of particular note is Ze et al. (2023), which, despite using multiple camera views to synthesize unseen views through neural rendering and extracting features from pre-trained models like StableDiffusion, shows advancement in connecting sparse with dense observations. Nonetheless, the processes of synthesizing and propagating unseen views demand significant effort. Moreover, these efforts are predominantly focused on simple manipulations using parallel grippers, with intricate dexterous manipulations largely remaining unaddressed.

Another pertinent work is by Karunratanakul et al. (2020), which employs a signed distance field to illustrate interactions between human hands and objects. Our work parallels Karunratanakul et al. (2020) in optimizing hand parameters from a 3D field. However, unlike their direct depiction of hand-object distances, we construct an implicit feature field to define energy functions for end-effector parameters, offering a novel approach to complex dexterous manipulations.

**Distilling 2D Features into 3D** The works of Zhi et al. (2021), Siddiqui et al. (2023), and Ren et al. (2022) demonstrate the lifting of semantic information from 2D segmentation networks to 3D, showing that averaging language embeddings over views can produce distinct 3D segmentations. Kobayashi et al. (2022) and Tschernezki et al. (2022) delve into integrating pixel-aligned image features from models like LSeg or DINO (Caron et al., 2021) into 3D Neural Radiance Fields (NeRF), highlighting their impact on manipulating 3D geometry. Further, Peng et al. (2023) and Kerr et al. (2023) explore distilling non-pixel-aligned image features, such as those from CLIP (Radford et al., 2021), into 3D scenes without the need for fine-tuning, yet their reliance on dense view acquisition for 3D feature extraction poses challenges for scenarios limited to sparse camera setups.

**Dexterous Grasping** Dexterous manipulation, central to advanced robotic applications, necessitates nuanced understanding and control, akin to human-like grasping capabilities (Salisbury & Craig, 1982; Dogar & Srinivasa, 2010; Andrews & Kry, 2013; Dafle et al., 2014; Kumar et al., 2016; Qi et al., 2023; Liu et al., 2022). The field prioritizes dexterous grasping due to its foundational role in hand-object interactions. Analytical approaches (Bai & Liu, 2014; Dogar & Srinivasa, 2010; Andrews & Kry, 2013) focus on direct modeling of hand and object dynamics, offering varying simplification levels, while recent strides in learning-based methods have introduced state (Chen et al., 2022; Christen et al., 2022; Andrychowicz et al., 2020; She et al., 2022; Wu et al., 2023b) and vision-based strategies (Mandikal & Grauman, 2021; 2022; Li et al., 2023a; Qin et al., 2023; Xu et al., 2023; Wan et al., 2023), targeting realistic scene comprehensions.

Despite their advances, these methods depend on large demonstration datasets for training, showing limited generalization beyond trained scenarios. Notably, Wei et al. (2023) proposes a grasp synthesis algorithm that generalizes within similar object shapes using minimal demonstrations, focusing on geometric and physical constraints for functional grasping. Our approach distinguishes itself by employing semantic visual features for dexterous manipulation, facilitating cross-category generalization from a singular demonstration, thus broadening the scope of generalization in dexterous grasping.

## 3 METHOD

Given a 3D point cloud $\mathbf{X}$, we aim to first construct a continuous feature field $\mathbf{f}(\cdot, \mathbf{X}) : \mathbb{R}^3 \rightarrow \mathbb{R}^C$ surrounding the scene, as illustrated in Fig. 2. This field provides semantic understandings for inter-scene correspondences, extending beyond geometric descriptors (Secs. 3.1 and 3.2). Utilizing a lightweight feature network, trained once on the source scene, enables direct adaptation to target scenes without additional fine-tuning. We then introduce a pruning method to enhance feature continuity, akin to the classical Hough voting (Qi et al., 2019). Following this, we employ the demonstrated end-effector pose to formulate an energy function between source and target scenes via the feature fields. This function facilitates the optimization of the end-effector pose in the target scene while conforming to physical constraints (Sec. 3.3), depicted in Fig. 3.

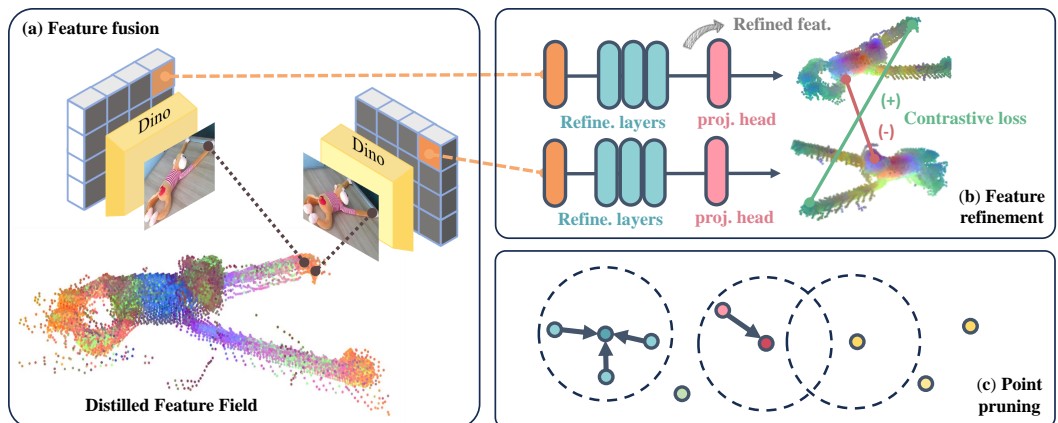

Figure 2: **Constructing sparse-view DFFs.** (a) Starting with the aggregation of DINO features, we form an initial 3D DFF. (b) Next, a lightweight network then refines these features, trained solely on a single demonstration and employing contrastive loss to improve field consistency. (c) Finally, a pruning algorithm assesses points through feature similarity in their vicinity. Points with minimal votes are eliminated.

## 3.1 3D FEATURE DISTILLATION

In contrast to previous works (Kerr et al., 2023; Rashid et al., 2023; Ze et al., 2023) that directly reconstruct continuous implicit feature fields alongside NeRF representations, our method first distills features onto discrete 3D points before propagating them into the surrounding space (Fig. 2a), akin to the approach of Kerbl et al. (2023). Formally, for any given point cloud feature set $\mathbf{F} = \{\mathbf{f}_i\}$, the feature $\mathbf{f}$ at a query point $\mathbf{q} \in \mathbb{R}^3$ is determined by

$$\mathbf{f} = \sum_{i=1}^{N} w_i \mathbf{f}_i, \quad \text{where} \quad w_i = \frac{1/\|\mathbf{q} - \mathbf{x}_i\|^2}{\sum_{j=1}^{N} 1/\|\mathbf{q} - \mathbf{x}_j\|^2}. \tag{1}$$

Consider a 3D scene observed by several cameras positioned around it, yielding a set of $K$ sparsely sampled RGBD scans. Each scan comprises a 2D image and a 2.5D point cloud $\mathbf{X}_k \in \mathbb{R}^{N_k \times 3}$, establishing 1 to 1 correspondences between image pixels and 3D points. Combining the point clouds of $K$ views produces a comprehensive 3D point cloud of the scene $\overline{\mathbf{X}} = \bigcup_k \mathbf{X}_k \in \mathbb{R}^{N \times 3}$.

As large vision models such as DINO (Caron et al., 2021; Oquab et al., 2023) have demonstrated emergent object correspondence properties even when trained without supervision, an intuitive approach is to directly apply DINO (Oquab et al., 2023) to RGB images and back-project to each point cloud $\mathbf{X}_k$ using the pixel-point correspondences, yielding per-point features $\mathbf{F}_k \in \mathbb{R}^{N_k \times C}$; integrating the features of all views results in the point features of the entire scene $\overline{\mathbf{F}} = \bigcup_k \mathbf{F}_k \in \mathbb{R}^{N \times C}$. However, DINO lacks strict multiview invariance, causing local feature inconsistencies.

Addressing the issue of local feature discrepancies, we devise a lightweight feature refinement network $\varphi$, consisting of a shallow per-point MLP, as depicted in Fig. 2b. This network can (i) be efficiently self-supervisedly trained on a single source scene, (ii) obtain high-quality, consistent feature, and (iii) directly apply in new scenes without any modification. The efficacy of this feature refinement is demonstrated and discussed in Sec. 4.3. Formally, given any pair of 2.5D point clouds $\mathbf{X}_k = \{\mathbf{x}_{kn}\}, \mathbf{X}_l = \{\mathbf{x}_{lm}\}$ from different views with per-point DINO features $\mathbf{f}_{kn}, \mathbf{f}_{lm}$, applying $\varphi$ yields the refined features $\mathbf{f}'_{kn} = \varphi(\mathbf{f}_{kn}), \mathbf{f}'_{lm} = \varphi(\mathbf{f}_{lm})$. The intuition is to ensure that neighboring features are similar and distant ones are distinct, following Xie et al. (2020).

Computationally, we use contrastive learning (Chen et al., 2020) to optimize the weights in $\varphi$. The refined features are passed through a projection head $g$ to obtain projected features $\mathbf{g}_{kn}, \mathbf{g}_{lm}$. The contrastive learning objective is defined by finding the overlapping region of the two views with pairs of points (distance ¡1cm). In each training iteration, a minibatch of $N$ pairs is randomly sampled as positive examples, with the other $2(N-1)$ possible pairs of unmatched points serving as negative examples. The contrastive loss for a positive pair of examples $(\mathbf{x}_{kn}, \mathbf{x}_{lm})$ is formulated as

$$l_{nm} = -\log \frac{\exp(\text{sim}(\mathbf{g}_{kn}, \mathbf{g}_{lm})/\tau)}{\sum_{i=1}^{2N} \mathbf{1}_{[i \neq n]} \exp(\text{sim}(\mathbf{g}_{kn}, \mathbf{g}_{li})/\tau)}, \tag{2}$$

with $\text{sim}(\mathbf{u}, \mathbf{v})$ as cosine similarity, $\mathbf{1}_{[i \neq n]}$ indicating $i \neq n$, and $\tau$ as a temperature parameter. After training on the source scene, $g$ is removed, retaining only $\varphi$ for novel scenes.

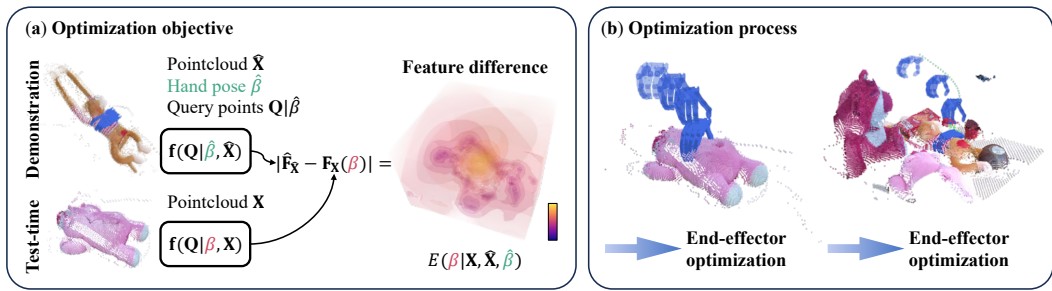

Figure 3: **End-effector optimization.** (a) We sample query points on the end-effector and compute their features using the learned 3D feature field. Minimizing the feature differences as an energy function facilitates the transfer of the end-effector pose from the source demonstration to the target manipulation. (b) The color gradient on the hand indicates the optimization steps from start to end.

## 3.2 POINT PRUNING

To improve feature consistency within the merged point cloud $\overline{\mathbf{X}} = \{\mathbf{x}_i\}$, we introduce a pruning mechanism akin to Hough voting (Xie et al., 2020), depicted in Fig. 2c. mechanism operates based on the refined point features $\mathbf{f}'$, focusing on the similarity of features among neighboring points.

Specifically, each point $\mathbf{x}_i$ is evaluated within a radius $r$. For every neighboring point $\mathbf{x}_j \in \mathcal{B}(\mathbf{x}_i, r)$, we assess the feature difference between $\mathbf{f}'_j$ and $\mathbf{f}'_i$. If the discrepancy $\|\mathbf{f}'_i - \mathbf{f}'_j\|$ falls below a predefined threshold $\delta$, then $\mathbf{x}_i$ secures a vote from $\mathbf{x}_j$. The voting is formalized as:

$$\mathcal{V}(\mathbf{x}_i) = \#\{\mathbf{x}_j \in \mathbb{B}(\mathbf{x}_i, r) : \|\mathbf{f}'_i - \mathbf{f}'_j\| < \delta\}, \tag{3}$$

leading to the exclusion of the bottom 20% of points with the least votes.

This pruning mechanism addresses potential discontinuities between adjacent points from different views, a common issue when integrating DINO features, which are generally continuous within the same image but may diverge across views. By pruning points that lack consensus across views, we enhance feature consistency and reliability across the point cloud, mitigating the impact of point cloud noise and DINO feature imperfections.

## 3.3 END-EFFECTOR OPTIMIZATION

For our dexterous hand end-effector, represented by the joint pose parameters $\beta$, we optimize these parameters in the target scene $\mathbf{X}$ using the feature fields derived from both the source demonstration scene $\hat{\mathbf{X}}$ with hand parameters $\hat{\beta}$, and the target scene; please refer to Fig. 3 for visualization.

Starting with randomly sampling $Q$ points on the hand surfaces in both the source and target manipulation, we generate query point sets $\mathbf{Q}|\hat{\beta}, \mathbf{Q}|\beta \in \mathbb{R}^{Q \times 3}$, conditioned on the hand parameters. Prioritizing the fingers for their crucial role in dexterous manipulation, we ensure a higher sampling density on them than on the palm. These query points are then processed through our learned 3D feature fields to extract feature sets $\mathbf{f}(\mathbf{Q}|\hat{\beta}, \hat{\mathbf{X}}), \mathbf{f}(\mathbf{Q}|\beta, \mathbf{X}) \in \mathbb{R}^{Q \times C}$. The objective is to minimize the feature disparity between the demonstration and target hand poses via an $l1$ loss, formulated as an energy function $E(\beta|\mathbf{X}, \hat{\mathbf{X}}, \hat{\beta})$ w.r.t. $\beta$:

$$E(\beta|\mathbf{X}, \hat{\mathbf{X}}, \hat{\beta}) = |\mathbf{f}(\mathbf{Q}|\hat{\beta}, \hat{\mathbf{X}}) - \mathbf{f}(\mathbf{Q}|\beta, \mathbf{X})|. \tag{4}$$

We integrate repulsion energy functions to prevent hand-object and self-penetrations, drawing from Wang et al. (2023) to ensure the action's physical viability. The inter-penetration and self-penetration energy functions are as follows:

$$E_{\text{pen}}(\beta|\mathbf{X}) = \sum_{\mathbf{x} \in \mathbf{X}} \mathbf{1}_{[\mathbf{x} \in \overline{Q}]} d(\mathbf{x}, \partial\overline{\mathbf{Q}}), \quad E_{\text{spen}}(\beta) = \sum_{\mathbf{p}, \mathbf{q} \in \mathbb{Q}} \mathbf{1}_{\mathbf{p} \neq \mathbf{q}} \max(\delta - d(\mathbf{p}, \mathbf{q}), 0). \tag{5}$$

Moreover, to avoid potential physical damage from extreme hand poses, a pose constraint $E_{\text{pose}}(\beta)$ penalizes out-of-limit hand pose. The overall optimization combines these terms:

$$E(\beta|\mathbf{X}, \hat{\mathbf{X}}, \hat{\beta}) + \lambda_{\text{pen}} E_{\text{pen}}(\beta|\mathbf{X}) + \lambda_{\text{spen}} E_{\text{spen}}(\beta) + \lambda_{\text{pose}} E_{\text{pose}}(\beta). \tag{6}$$

In our implementation, we set $\lambda_{\text{pen}} = 10^{-1}, \lambda_{\text{spen}} = 10^{-2}, \lambda_{\text{pose}} = 10^{-2}$.

# 4 EXPERIMENTS

We evaluate our model through real-world experiments with a robot hand, opting for direct assessment in real-world settings to leverage the superior stability of large vision models like DINO (Caron et al., 2021; Oquab et al., 2023) on real images over synthetic ones.

**Environment**    Our experimental setup features a Shadow Dexterous Hand, which has 24 Degrees of Freedoms (DoFs). We restrict the 2 DoFs at the wrist, focusing the optimization on the remaining 22 DoFs. This hand is mounted on a UR10e arm, adding 6 DoFs, enabling it to reach any point on tabletops sized either 1m×1.2m or 1m×1m. Precautions are taken to prevent the dexterous hand from contacting the table surface directly, and the range of motion is limited by the hand's elbow.

For RGBD scans, we use four Azure Kinect DK sensors, positioned at each corner of the table and aimed towards its center, ensuring comprehensive capture of the scene. These sensors are pre-calibrated and fixed at specific heights. Post-processing is applied to the captured scans to remove background elements.

In single-object experiments, the object's point cloud is segmented using SAM (Kirillov et al., 2023). For multi-object scenarios, we apply physical constraints to limit the experimental area to the tabletop and employ RANSAC (Fischler & Bolles, 1981) to exclude the table surface from the analysis.

**Tasks and evaluation**    Our method undergoes a quantitative evaluation centered on the grasping of both rigid and deformable objects, with grasping providing a straightforward measure of success or failure. Initially, for each trial, a demonstration is set up within a virtual environment on an object scan by manually positioning a dexterous hand model on the object's point cloud using MeshLab. After this setup, our feature network takes 20000 iterations for adaptation, roughly 300 seconds using a single NVIDIA GeForce RTX 3090. Once trained, the network is applied unchanged to different real-world scenes to optimize the hand pose for 300 iterations, roughly 20 seconds using a single NVIDIA GeForce RTX 3090.

During testing, object placements are varied within the hand's reach, and our approach is tested 10 times to determine its success rate. Each trial begins with deriving an initial grasping pose by our end-effector optimization process, succeeded by a predetermined lifting motion—either directly upwards or a mix of upward and backward movements, depending on the object's physical properties. A trial is deemed successful if the object is securely lifted from the table without being dropped.

**Baselines**    We compare our approach against a naive DFF baseline, where DINO image features are directly back-projected onto the point cloud following the method described in Peng et al. (2023), with interpolation then applied to populate the 3D space. This comparison leverages the nascent field of one-shot learning for dexterous manipulation, using identical end-effector optimization processes between our method and the baseline for fair comparison. In scenarios with rigid objects, we also benchmark against UniDexGrasp++ (Wan et al., 2023). Due to UniDexGrasp++'s vision model instability with our real-world, noisy point clouds, we evaluate its state-based model in simulations, conducting 100 trials per setup to gauge success rates.

## 4.1 RIGID OBJECTS

Our assessment of rigid object grasping is to validate the versatility of our method across different poses, shapes, and object categories. The evaluation focuses on specific setups:

- **Box:** A demonstration with the Cheez-It box from the YCB dataset (Calli et al., 2015b; 2017; 2015a) (ID=3) is utilized for initial evaluation. The testing extends to this same box and another cracker box, highlighting variations in geometry and appearance.
- **Drill:** The demonstration involves a functional grasp of the Drill from the YCB dataset (ID=35) by its handle. The testing includes the same drill in various poses.
- **Bowl:** For this setup, three 3D-printed bowls are used. The demonstration shows a grasp by the rim of Bowl1, with subsequent tests on all three bowls, each presented in different poses, including a cat-shaped bowl with unique shape and complex geometry.
- **Bowl → Mug:** Demonstrating the method's generalization ability, the grasp learned on Bowl1 is transferred to three distinct 3D-printed Mugs. This setup tests the method's categorical generalization from Bowls to Mugs, which have related functionalities but distinct geometries.

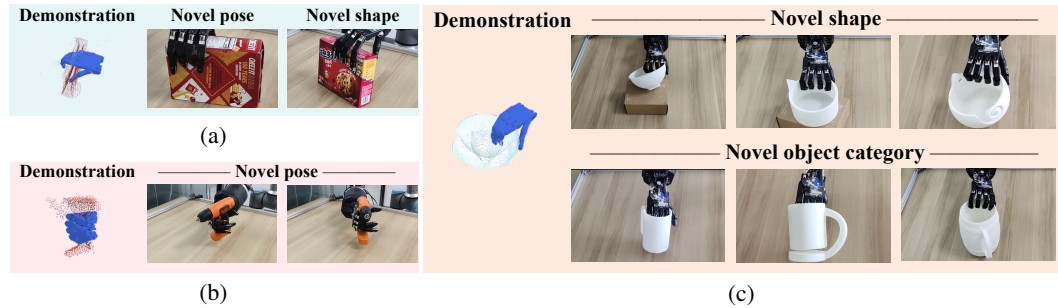

Figure 4: **Qualitative results on rigid objects grasping.** Each panel illustrates the initial grasping pose, determined via our end-effector optimization, followed by a frame capturing the successful lift-off of the target object. (a) Grasping Box1 and transferring the skill to Boxes in new poses, including a distinct box Box2. (b) A functional grasp of a drill by its handle. (c) Transferring the learned grasp on Bowl1 to bowls with varied shapes (top row) and cross-category generalization to Mugs (bottom row).

Tab. 1 outlines the success rates achieved by our method versus the baselines in rigid object grasping tasks. Our approach consistently surpasses the baseline performances across all configurations. While the baselines show promise in simpler scenarios—especially when the manipulation target closely matches the demonstrated object—their performance significantly drops as the source and target objects become more disparate, notably in transitions like from Bowl1 to CatBowl or Bowl1 to Mugs. Conversely, our method exhibits strong success rates even in these complex transitions.

Table 1: **Success rates on rigid object grasping.** Given the instability of the vision-based model of UniDexGrasp++ (Wan et al., 2023) with our noisy point clouds, we evaluate its state-based model within simulation environments on virtually replicated objects. * denotes experiments conducted in simulation.

| Demo. | Box1 | | Drill | | | | Bowl1 | | |
|---|---|---|---|---|---|---|---|---|---|
| Target | Box1 | Box2 | Drill | Bowl1 | Bowl2 | CatBowl | Mug | FloatingMug | BeerBarrel |
| UniDexGrasp++* | 7.7% | - | 66.9% | 37.7% | 31.9% | 26.3% | 24.7% | 25.5% | 6.2% |
| DFF | 90% | 0% | 100% | 100% | 0% | 30% | 0% | 20% | 10% |
| Ours | 100% | 100% | 100% | 100% | 80% | 60% | 80% | 40% | 90% |

Fig. 4 showcases our qualitative results. Specifically, the Boxes and Drill from the YCB dataset are depicted in Figs. 4a and 4b, demonstrating our method's ability to generalize across rigid transformations of object poses. The transfer from Box1 to Box2 is also illustrated in Fig. 4a, showcasing our method's adaptability to variations in geometry and appearance. In Fig. 4c, we show the learned grasping from a demonstration on Bowl1, extending its applicability to diverse bowls, including the geometrically complex CatBowl (top row), as well as the cross-category generalization from Bowls to Mugs (bottom row). Refer to *Supplementary Video* for additional qualitative results.

## 4.2 DEFORMABLE OBJECTS

Our evaluation extends to deformable objects to showcase the method's adaptability across various deformations, object types, and scene contexts. The setups include:

- **SmallBear, BigBear, Monkey:** This category involves three plush toys—two Bears of different sizes, and a Monkey characterized by its flexibility and elongated limbs. Each toy is demonstrated in a specific pose, with subsequent evaluations exploring diverse poses and levels of deformation.
- **Monkey ↔ SmallBear:** Additionally, we explore the method's capability to transfer grasping knowledge between the Monkey and the SmallBear. This task underscores the challenges of adapting between different objects and their respective deformations.
- **Monkey in Context:** A complex scene setup features the Monkey amidst various background items, with each trial randomizing object placements. This environment tests the method's performance in scenarios where the Monkey may be partially obscured by surrounding objects. In comparison, the grasp demonstration is performed with the Monkey isolated from these complicating factors.

The success rates for the deformable objects are detailed in Tab. 2. Our method markedly outperforms the baseline, demonstrating exceptional success rates, especially in scenarios that re-

Table 2: **Success rates on deformable object grasping.**

| Demo. | Monkey | | | BigBear | SmallBear | |
|---|---|---|---|---|---|---|
| Target | Monkey | MonkeyScene | SmallBear | BigBear | SmallBear | Monkey |
| DFF | 90% | 40% | 0% | 20% | 90% | 0% |
| Ours | 100% | 100% | 60% | 80% | 90% | 50% |

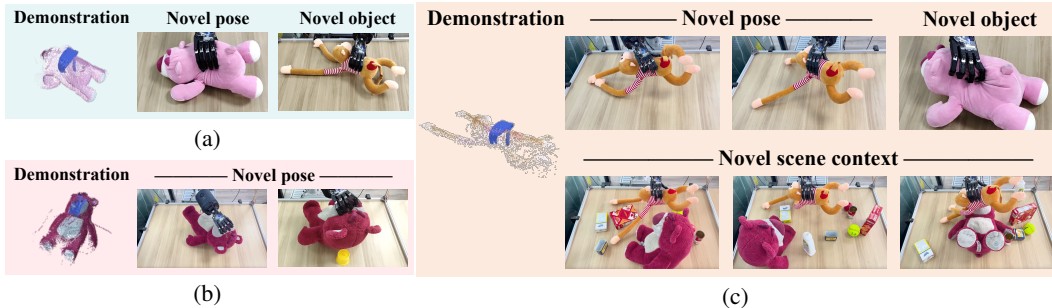

Figure 5: **Qualitative results on deformable objects grasping.** For each successful grasp, we show the initial grasping pose and a frame demonstrating the successful lift of the object off the table. (a) Learning to grasp SmallBear and transferring this skill to various poses and to the Monkey. (b) Learning to grasp BigBear by the nose is challenging due to its small nose. (c) Learning to grasp the Monkey, showcasing adaptability to significant deformations and transfers to SmallBear. Additionally, a challenging scenario is presented where the Monkey is surrounded by multiple objects, showing the capability to handle interactions and occlusions.

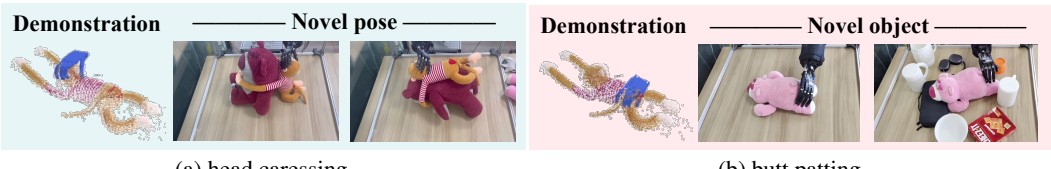

Figure 6: **Pet toy animals.** (a) Head caressing is transferred from a single, lying Monkey to a scene with the Monkey hugging the BigBear, exemplifying the method's adaptability to varying scene compositions and interactions. (b) Butt patting is transferred from the Monkey to the SmallBear, whether the SmallBear is alone or in different scene contexts, underlining the method's versatility across various scenarios and object interactions.

quire generalization. These find-
ings are consistent with the outcomes observed in the evaluations involving rigid objects.

Fig. 5 showcases our qualitative results in dealing with deformable objects. In Fig. 5a, we demonstrate the grasp of a SmallBear toy by its body, illustrating our method's capacity for generalization across different poses and to a Monkey toy, thereby underscoring the versatility of our approach. Fig. 5b shows a unique instance of grasping BigBear by its nose, reflecting our method's adaptability to the physical characteristics of diverse objects. Fig. 5c focuses on grasping the Monkey toy, emphasizing the method's flexibility with extensive deformations and its ability to generalize from the Monkey to the SmallBear, across varying object forms and structures. Additionally, Fig. 5c presents a complex test setup with the Monkey amidst a dynamic scene, requiring not just target object identification but also an intricate understanding of its pose, deformation, and relationships with surrounding objects, even in cases of occlusion. These instances highlight our method's robustness and flexibility in handling a range of challenging scenarios, showcasing its potential for real-world applications. Further qualitative results and discussion can be found in the *Supplementary Video*.

**Beyond grasping** Our framework's capabilities extend beyond simple grasping to encompass a variety of hand-object interactions. Fig. 6 depicts two distinct scenarios involving interactions with toy animals, demonstrating our method's adaptability. In Fig. 6a, the interaction entails caressing the head of a Monkey toy in various poses. The initial demonstration features the Monkey lying flat with its arms outstretched, while in subsequent tests, it adapts to hugging a BigBear, seamlessly adjusting to this new context. Fig. 6b focuses on patting the butts of the toys. This action, demonstrated on the Monkey, is successfully generalized to the SmallBear in various settings, highlighting our method's ability to adapt to different scenarios and object interactions.

## 4.3 ABLATION STUDIES

**Ablations on feature refinement** Our ablation on the feature refinement network, as discussed in Sec. 3.1, is visualized in Fig. 7. This network plays a crucial role in enhancing the feature field's consistency. The comparison highlights the energy fields used for end-effector pose optimization, showcasing the advantages of incorporating the refinement network. With refinement, we observe a focused distribution of low-energy values at positions matching the hand demonstration, indicating

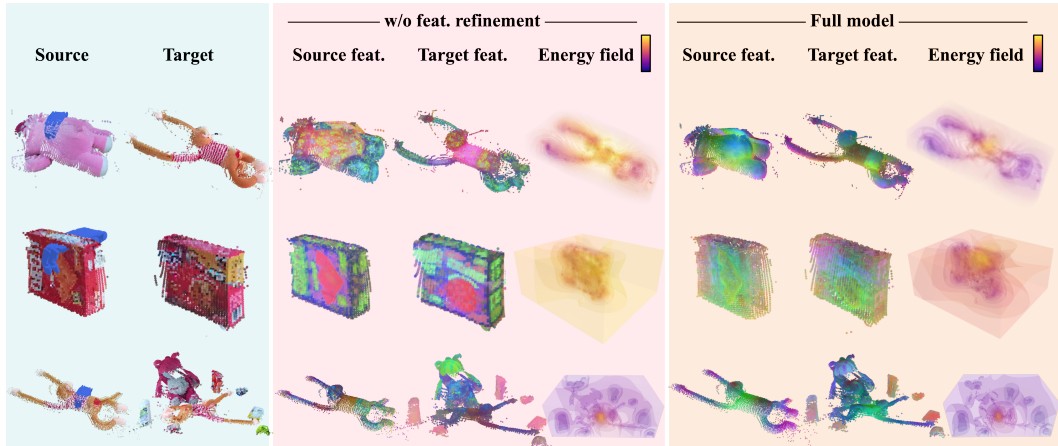

Figure 7: **Ablation study on feature refinement.** Given the source and target objects (left), we compare the source and target features, the energy fields between a model without the feature refinement network (middle), and the full model incorporating it (right). Point cloud features are visualized using RGB coloring, derived by applying PCA to reduce the features to three components. The energy fields, calculated based on the feature differences between the source and target scenes, are illustrated to reflect differences: yellow areas signify smaller differences and thus higher similarity, whereas purple areas indicate larger differences, emphasizing feature dissimilarity.

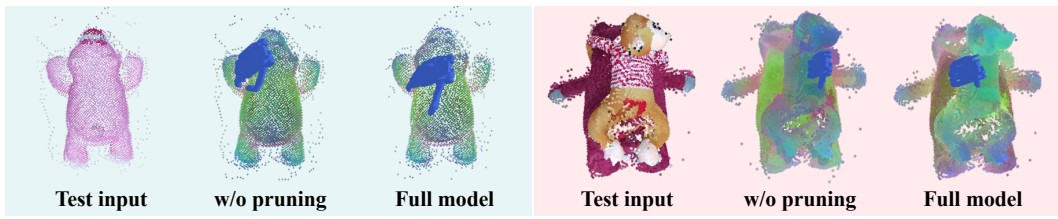

Figure 8: **Ablations on point pruning.** We show the end-effector grasping poses in the test scene (left), comparing the results with (middle) and without (right) point pruning; the demonstration is omitted for simplicity. Features within the point clouds are visualized in RGB, achieved by reducing feature dimensions to three components using PCA. The optimized hand positions are highlighted in blue.

better feature consistency. Conversely, without refinement, the energy distribution appears more scattered, lacking precise low-energy zones.

**Ablations on point pruning** Fig. 8 qualitatively demonstrates the impact of point pruning (Sec. 3.2) using two examples. This procedure not only eliminates outlier points but also boosts the local feature consistency, significantly enhancing the optimization stability and the accuracy of end-effector poses.

## 5 CONCLUSIONS

In this study, we tackled the challenge of one-shot learning for dexterous manipulations, harnessing semantic correspondences from pre-trained vision models to enhance robotic grasping capabilities. By distilling sparse-view RGBD observations into a consistent 3D feature field, we developed a method that significantly advances the adaptability and generalization of robotic manipulators across various objects and scenes. Our approach not only demonstrated robust generalization to new object poses and categories in real-world settings but also showcased the potential for practical applications in dynamic environments. Future work will explore the integration of additional sensory inputs, such as tactile feedback, for further improving the precision and reliability of robotic grasping.

**Acknowledgment** The authors thank NVIDIA for their support of GPUs and hardware. C. Deng, Y. You, and L. Guibas are supported in part by the Toyota Research Institute University 2.0 Program and a Vannevar Bush Faculty Fellowship. Q. Wang, H. Zhang, and Y. Zhu are supported in part by the National Science and Technology Major Project (2022ZD0114900) and the Beijing Nova Program.

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
