# OpenReview forum: "SparseDFF: Sparse-View Feature Distillation for One-Shot Dexterous Manipulation"
_ICLR.cc/2024/Conference — ICLR 2024 poster_

### Official Review · Reviewer_hwb9 · 2023-10-29

**Soundness:** 3 good
**Presentation:** 3 good
**Contribution:** 3 good
**Rating:** 6
**Confidence:** 3

**Summary:**

This paper presents an innovative method to obtain view-consistent 3D DFFs from sparse RGBD data, enabling one-shot learning of complex manipulations that can be adapted to unfamiliar settings. The key contribution of SparseDFF comprises a lightweight feature refinement network, optimized using a contrastive loss applied to pairs of views after projecting image features onto the 3D point cloud. Furthermore, by establishing consistent feature fields in both the source and target scenes, they design an energy function that simplifies the process of minimizing feature differences with respect to the end-effector parameters between the demonstration and the target manipulation.

**Strengths:**

This paper presents a captivating approach to 3D feature learning, involving the creation of a point-cloud-based 3D representation and the utilization of the DINO feature extractor. This 3D representation based on point clouds can enable one-shot dexterous manipulation. As demonstrated by the experimental results, the proposed method exhibits robust performance across various settings.

**Weaknesses:**

In light of the experimental findings presented in this paper, it is respectfully suggested that the method described may not be particularly captivating. For a more comprehensive critique, kindly refer to the Questions section.

**Questions:**

### Question 1:
In the contrast learning process, a distance of 1cm is set as the threshold to distinguish between similar and dissimilar parts. Can the authors provide clarification on how they precisely define this distance?
### Question 2:
Is distance truly an effective criterion for distinguishing between similarity and dissimilarity?
### Question 3:
Is there a typographical error in Equation (3)? Why does it contain both an equation symbol and an inequality symbol?
### Question 4:
Additionally, is the pruning process defined in Equation (3) considered reasonable?
### Question 5:
Could this pruning process potentially lead to the removal of critical edge information?

---

> ### Author Response · Authors · 2023-11-15
>
> Thanks for your feedback! Here are our responses to your questions and comments and we hope that they could address your concerns:
>
> **Threshold distance choice for the contrastive loss.**
>
> Following the PointContrast paper, let $X_1, X_2$ be two partial point clouds from two views. We sample point pairs from them $(x_{1i}, x_{2i})$. We say that it is a positive pair if $||x_{1i} - x_{2i}||$ < 1cm and a negative pair elsewise. The positive and negative pairs are then plugged into the PointNCE loss defined in Equation 2.
>
> Here we note that, although this threshold is hard, the contrastive loss itself is a soft loss for two reasons: (a) the weighting in the soft PointNCE loss, and (b) the balanced sampling rate between positive and negative pairs.
>
> The specific threshold distance is decided based on the object scale and the overall framework is not too sensitive to this hyperparameter within a reasonable range.  We will add more clarifications to the paper.
>
> **Is distance truly an effective criterion for distinguishing between similarity and dissimilarity?**
>
> For the feature similarities, we majorly follow prior works on their choices of metrics. Specifically:
> - We choose the cosine similarity for contrastive learning, as the InfoNCE loss with theoretical foundations (ref point contrast)
> - We choose L1-distance for computing the energy function during optimization, the same as in prior works such as NDF. Here the dominant factor for us to make the choice is not the criteria for similarity, but the optimization curve/landscape. We tried L1-distance, L2-distance, and cosine similarity here and observed that optimizing the hand pose with L1-distance gives the best quality and stability.
>
> **Symbols in Equation (3).**
>
> The # means the cardinality of the set defined in the brackets {...}, and the inequality $||f’_i - f’_j|| < \delta$ defines the region of the set. There are multiple notation systems for sets and we picked the one we’re used to. We apologize for the confusion and will add further clarifications to the paper.
>
> **Is the pruning process defined in Equation (3) considered reasonable?**
>
> The overall intuition of keeping 80% percent of points is that: (a) we want to discard the points with the worst features (here we recognize “bad” as lower feature consistency within local regions), but (b) we don’t want to discard way too many points to maintain the overall shape of the object/scene.
>
> **Could this pruning process potentially lead to the removal of critical edge information?**
>
> This is indeed a very good question. We think this should not be a big concern, as the pruning is majorly based on the features instead of point cloud geometry, and it is after the feature refinement. Sharp structures such as edges and corners usually easier for DINO to extract stable and coherent features. In fact, based on our observation, if we gradually decrease the percentage of points remaining (from 80% to 5%), the points leaving till the last are mostly the edge points which are more identifiable by the feature extractor. For 80% of points remaining, both edge and face points are kept and their distribution is fairly even across the object. We will add these discussions to the paper.

---

> ### Author Response · Authors · 2023-11-21
>
> Thank you once again for your valuable comments on our work. We have already provided our response to your suggestions. As the deadline for discussion is approaching, we kindly request you to promptly provide any further questions or concerns you may have.

---

> > ### Comment · Reviewer_hwb9 · 2023-11-21
> > **Official Comments**
> >
> > Thank you very much for your reply, which effectively addresses my issue.

---

### Official Review · Reviewer_sPkV · 2023-10-29

**Soundness:** 3 good
**Presentation:** 4 excellent
**Contribution:** 3 good
**Rating:** 6
**Confidence:** 4

**Summary:**

This work applies 3D feature fields for manipulation. The key contribution lies in introducing a sparse view setting, where unlike previous works that use dense RGB views, this work uses sparse RGB-D views. A point-based sparse 3D feature field construction method is introduced to improve the 3D information aggregation quality and the grasping task performance.

**Strengths:**

The introduction of sparse RGBD camera setting.
The method of sparse DFF to reconstruct 3D feature fields from sparse RGBD inputs.
Reasonable experiment design and analysis.

**Weaknesses:**

There are existing sparse-view NeRF methods (e.g., [1,2]) that applies similar ideas as the sparse DFF, some of which are not extremely hard to apply to the normal DFF (e.g,, [1]). It is fairer to allow baselines to also utilize the depth information introduced in this work (e.g., introducing depth supervision similar as [1] in DFF).

[1] Depth-supervised NeRF: Fewer Views and Faster Training for Free
[2] MVSNeRF: Fast Generalizable Radiance Field Reconstruction from Multi-View Stereo

**Questions:**

N.A.

---

> ### Author Response · Authors · 2023-11-15
>
> Thanks for your feedback! Here are our responses to your questions and comments and we hope that they could address your concerns:
>
> **Sparse-view or depth-guided NeRF methods.**
>
> In our baseline comparison, we also used the depth for DFF (we apologize for not making it clear and will add clarifications to the paper). Therefore, we remove the difficulty of reconstructing the 3D geometry in both our method and the baselines and majorly focus on extracting the latent features for manipulation.
>
> Specifically, we want to address that, 3D consistency itself also exists in DFF/NeRF which directly averages the features from different views (because only the mean is considered and the variance is omitted). However, such view averaging with inconsistent 2D features can result in low-quality 3D features – which is the message we want to convey in this paper.
>
>
> **Fairer to utilize depth in the baseline.**
>
> - As mentioned above, for the DFF baseline, we are also using the scanned depth same as in our method for a fair comparison. We apologize for not clarifying this in the paper.
> - For UniDexGrasp++, we are using their state-based model which takes the ground-truth 3D geometry as input.
>
> We will add further clarifications to the paper.

---

> > ### Comment · Reviewer_sPkV · 2023-11-22
> >
> > Thanks for the response, though I am not sure whether providing the depth information to the baselines without proper incorporation of the techniques in [1] or [2] is the best practice, I understand that conducting such extra experiments would be impractical during rebuttal. I would keep the score the same (slightly above the acceptance threshold) and suggest the authors to  consider this point further in the camera ready, i.e., either explain clearly why techniques like [1] or [2] do not apply to the baseline, or conduct extra experiments.

---

> > > ### Author Response · Authors · 2023-11-22
> > >
> > > Thanks for your reply, and for keeping your positive rating!
> > >
> > > Though not providing additional experimental results, we'd like to provide a bit more intuitions and observations here, and we hope that they could better address your concerns: In our early experiments, we've tried Gaussian splatting, which also uses depth (SfM points for growing the Gaussians in the original paper), and the results were not very good. In general, our observation is that the main bottleneck for obtaining a good feature field is not in the reconstruction (as the depth from 4 views already give decent geometry), but in the per-view image features themselves -- which is also why we introduce the contrastive feature refinement module.
> > >
> > > We will add more content to the paper to address these points.

---

> ### Author Response · Authors · 2023-11-21
>
> Thank you once again for your valuable comments on our work. We have already provided our response to your suggestions. As the deadline for discussion is approaching, we kindly request you to promptly provide any further questions or concerns you may have.

---

### Official Review · Reviewer_dYVL · 2023-11-01

**Soundness:** 3 good
**Presentation:** 2 fair
**Contribution:** 3 good
**Rating:** 6
**Confidence:** 3

**Summary:**

To mitigate the dense view requirement in the distilled feature field (DFF) for the application of one-shot dexterous manipulation, the authors introduce sparseDFF, which utilizes a sparse collection of RGB-D scans of a scene. Feature points are reprojected using depth, followed by a feature refinement process on these reprojections. Contrastive loss and point pruning are employed to enhance feature consistency within each local neighborhood, and an energy function is formulated to aid in reducing feature discrepancies. Performance on grasping benchmarks demonstrate the proposed methods surpasses DFF while significantly outperforms UniDexGrasp++.

**Strengths:**

- The necessity of as few as 4 views for transferring manipulation skills is noteworthy, as this method can be readily generalized to novel scenes.
- Point feature refinement, the minimization of feature discrepancies using an energy function, and point pruning are specifically designed for applications with RGB-D scans as input.
- Utilizing DINO feature distillation for diverse downstream grasping tasks markedly outperforms the previous baseline (DFF) and UniDexGrasp++.

**Weaknesses:**

**[Clearance]**
- The authors are encouraged to establish connections regarding why distilling DINO features is advantageous and elucidate its applications in downstream tasks.
- The definition of **one-shot** should be explained in the manuscript (abstract or introduction).
- The input should accurately be described as "multi-view RGB-D scans" rather than "Given a 3D point cloud X", in the method section. And the dimension of the variable should be added.
- Regarding the motivation of using DINO, while the authors have highlighted, "This field offers semantic understandings for inter-scene correspondences that transcend geometric descriptors", how does it contribute specifically to image matching deep models like LOFTR?

**[Method]**
The authors propose "discard the 20% of points that accumulate the fewest votes.".How was this hyper-parameter for the pruning ratio determined? Were multiple-stage or iterative pruning strategies considered?

**[Experiments]**
- How did the inference performance compare with baseline methods?
- Was depth information also utilized by DFF?

**Questions:**

See the raised questions in Weaknesses.

---

> ### Author Response · Authors · 2023-11-15
>
> Thanks for your feedback! Here are our responses to your questions and comments and we hope that they could address your concerns:
>
> **Elucidating the advantages of distilling DINO features in the downstream tasks.**
>
> The features from large vision models such as DINO provide semantics-aware correspondences across different instances/categories, allowing one-shot/few-shot transfer of manipulation policies. However, the 2D image features are only sufficient for simple end-effectors such as parallel grippers operating on single points in the space. For dexterous hands with complex geometry, we need a 3D feature field to decide the parameters of all the finger joints, and that’s why we need such 2D-to-3D distillation. We will add more explanations and discussions to the paper.
>
> **The definition of one-shot should be explained in the manuscript (abstract or introduction).**
>
> Here the “one-shot” means that given one demonstration on the source object, we seamlessly transfer learned manipulations to novel scenes, accommodating variations in object poses, deformations, scene contexts, and even object categories. We abridge the wording a bit to avoid making the title too long. We will add more clarifications to the paper.
>
> **The input format and the dimension of the variables.**
>
> The 3D point cloud is obtained by merging the depth scans from the 4 views. We will add clarifications to the paper.
>
> **Motivation for using DINO and other possible choices of image-matching deep models like LOFTR.**
>
> Our framework is not specifically curated for DINO features and would be potentially compatible with any image feature extractors. We adopt DINOv2 here as they’ve shown strong cross-instance and cross-category correspondences. LOFTR seems to work mostly on matching two identical objects in different images. We will add the discussions of other image-matching frameworks like LOFTR to the paper.
>
> **Hyper-parameter for point pruning pruning. Multiple-stage or iterative pruning strategies.**
>
> We have tried different hyperparameters for the point pruning and within a certain range (70%-80% points left) it offers consistently good results. The overall intuition of choosing the hyperparameters is that: (a) we want to discard the points with the worst features, but (b) we don’t want to discard way too many points to maintain the overall shape of the object/scene.
>
> Muti-stage or iterative pruning would be an interesting future direction to study and thanks for pointing it to us.
>
> **How did the inference performance compare with baseline methods?**
>
> Our inference time is roughly the same as DFF, and both DFF and UniDexGrasp++ require iterative inference (RL or optimization) on the hand pose. This is because, for hand-like end-effectors with complex shapes, a variety of geometric and physical factors (e.g. contacts, penetrations, force closures, etc) need to be carefully decided for a successful manipulation (grasp). On the other hand, one-step forward prediction methods are prevalent for parallel grippers because they operate on single 3D points and can apply infinite forces.
>
> **Was depth information also utilized by DFF?**
>
> Yes, depth information is also used in DFF for a fair comparison. We apologize for not clarifying this in the paper.
>
> What we want to address in the paper is that, even with the scanned 3D geometry guiding the feature aggregation, the feature incoherency between different views, if not properly tackled, will lead to a lower-quality 3D feature field, causing a great drop in the success rate.

---

> > ### Author Response · Authors · 2023-11-21
> >
> > Thank you once again for your valuable comments on our work. We have already provided our response to your suggestions. As the deadline for discussion is approaching, we kindly request you to promptly provide any further questions or concerns you may have.

---

### Official Review · Reviewer_Txdk · 2023-11-02

**Soundness:** 3 good
**Presentation:** 3 good
**Contribution:** 3 good
**Rating:** 6
**Confidence:** 4

**Summary:**

The broader goal of the paper is to enable the transfer of robotic dexterous grasps from one object to a similar object (which can happen with understanding the inherent similarities of the 3D shape instances despite the variations in appearances, poses, or categories). More precisely, given a source scene, source hand-object grasp and a target scene, what could be the target grasps.
To do so, they leverage 2D vision models (namely DINO) to learn features in 2D space and then distill or back-project those features in 3D. The similarly of features in 3D space allows transfer of grasps from one object to another.
Compared to prior works, which leverage dense multi-view images to infer dense 2D features for each three 3D point and simply average the multi-view features, the paper operates in sparse view setting. Rather than simply out the point features from multiple view, the paper paper learns a feature refinement network on each point features and defines a contrastive learning approach to bring points closer to each in 3D other more closer in feature space and points farther from each other in 3D space more farther in feature space.
Finally, leveraging the projected and refined 3D features from sparse views, the paper performs the task mentioned in first bullet, mapping source grasp (on sourc e scene) to target scene.

**Strengths:**

Writing is good, and the paper is easy to follow.
The motivation of creating a generalized robotic manipulator and leveraging 3D priors for that, seems exciting and promising.

**Weaknesses:**

[Novelty Issue] Lack of any exciting factor or in some sense novelty: The difference from prior work in DFF (distill feature field) is that the paper operates in sparse view setting, where simple fusion of features from multiple views doesn’t perform best. To overcome this loss of views from prior work, they refine the point features baed on the insight that points close in 3D, should have similar features. This seems a very natural and obvious technique of pruning or refining points which have incorrect feature consistency w.r.t 3D consistency.

[Assumptions on GT depth and camera pose] Secondly, following up on the novelty part, the paper makes the assumption of having GT depth maps and also GT camera poses. Errors in either of them will make the above refinement step tricky.

[Additional experimental comparison] Experimental Comparison against baselines like (Neural descriptor fields and follow-up) where the goal is similar but rather than using the large vision model, an object category specific feature descriptor is learned.

[Less Relevant] Newer papers like Lseg, Conceptfusion have projected LLM/ vision-LLM features on the 3D scenes, comments on using them as feature backbones will be appreciated.

[Less Relevant] Papers learning joint hand object poses, leaning visual affordances from images, also seem to be relevant related works, comparison against them in related work would be appreciated (Affordance Diffusion: Synthesizing Hand-Object Interactions, papers on hand-object interaction: HOI etc).

**Questions:**

I would like authors to address the points raised in weaknesses section.
1. What happens when there is noise in pose and/or depth? How will that impact refinement module?
2. How does the method compare with baselines like neural descriptor fields and follow up works?

---

> ### Author Response · Authors · 2023-11-15
>
> Thanks for your feedback! Here are our responses to your questions and comments and we hope that they could address your concerns:
>
> **Novelty. It is natural that feature refinement induces 3D consistency.**
>
> Specifically, we want to address that, 3D consistency itself also exists in vanilla DFF/NeRF without any refinement that directly averages the features from different views (because only the mean is considered and the variance is omitted). However, such view averaging with inconsistent 2D features can result in low-quality 3D features – which is the message we want to convey in this paper.
>
> We agree that feature refinement w.r.t. 3D consistency gives better 3D consistency is not surprising. But what is not obvious is that feature refinement w.r.t. 3D consistency gives better cross-instance correspondences.
>
> Moreover, most contrastive feature learning (such as PointContrast) show their generalization after training on a large dataset of pairwise samples, but here we show that even only overfitted to the 4 views of the training scene with a few minutes of optimization, our refinement network naturally generalizes to novel scenes.
>
> Application-wise, compared to parallel grippers which operate on one single 3D point and can apply infinitely large force, dexterous hands require more accuracy in the end-effector pose to, contact the object surface, wrap the object, and get the forced closure. This makes a lot of methods for parallel grippers not directly applicable to dexterous hands. In such a scenario, fine-grained properties of the feature field become more important.
>
> **Noise in depth and camera pose.**
>
> All experiments for our method and the DFF baseline are on real robots, and the depth and camera poses are all from real data with scan errors and calibration errors. Only the UniDexGrasp++ baseline is run in the simulation environment with ground-truth 3D geometry (as we’re running their state-based model).
>
> The sim2real gap is a long-standing problem in robotics, and it is particularly severe for dexterous hands compared to simpler end-effectors such as parallel grippers. In this work, we show our results on real robots to demonstrate our robustness towards this sim2real gap.
>
> **Additional experimental comparison with NDF.**
>
> Here we provide additional comparisons with NDF. As we temporarily don’t have access to the hardware, we conduct the experiments in simulation environments (IsaacGym) with the object scans from the experiments in the paper.
>
> The success rates are:
>
> ————————————————————
>
> Source: Mug2, Target: Mug2 (same object)
> - NDF: 8/9, Ours: 9/9
>
> Source: Mug2, Target: BeerBarrel (same category)
> - NDF: 6/10, Ours: 9/10
>
> Source: Mug2, Target: CatBowl (cross category)
> - NDF: 0/10, Ours: 7/10
>
> ————————————————————
>
> (*There are only 9 test samples for Mug2 because one Mug2 scan is used for providing the demonstration. We will update the results as well as conduct real-robot comparisons once we get access to the hardware again.)
>
> In addition, we also address the following limitations of NDF:
> NDF needs category-specific training and it takes a lot of time.
> NDF is specifically designed for category-level rigid object manipulation. It is not applicable to other scenarios such as deformable objects or cross-category generalizations.
> As a pure geometry-based method, NDF is highly sensitive to geometric variations (such as the BeerBarrel and CatBowl in the experiment here).
> Following the previous question on scan noises, our method actually shows much better noise stability than NDF in real-world experiments.
>
> **Other LLM/VLM backbones for feature extractions.**
>
> We choose the latest DINOv2 here as they demonstrated strong cross-instance/category correspondences on images, while some other VLMs are more focused on language queries (such as open-vocab query/segmentation). But we agree that exploring different VLM backbones, especially studying their view consistency and 3D awareness, would be an interesting future work. We will also add more discussions to the paper.
>
> **Other related works on learning joint hand-object poses or learning visual affordances from images.**.
>
> Thanks for providing these references. We will add the discussions to our paper.

---

> ### Author Response · Authors · 2023-11-21
>
> Thank you once again for your valuable comments on our work. We have already provided our response to your suggestions. As the deadline for discussion is approaching, we kindly request you to promptly provide any further questions or concerns you may have.

---

> > ### Comment · Reviewer_Txdk · 2023-11-21
> >
> > I thank the authors for addressing my concerns. I have updated my rating from 5 to 6.

---

### Meta-Review · Area_Chair_nHH1 · 2023-12-09

**Metareview:**

This paper introduces a novel method, SparseDFF, for creating 3D Distilled Feature Fields from sparse RGBD observations, enabling robots to learn and transfer dexterous manipulations to novel scenes, demonstrating robust generalization in various real-world scenarios. The reviewers initially have some reservations on the papers but they are all addressed by the rebuttal. All the reviewers agree to accept the paper and the AC agrees with the reviewers on acceptance.

**Justification For Why Not Higher Score:**

The problem the paper tries to solve and the proposed approach is relatively standard. There does not seem to be a lot of excitement among the reviewers towards this paper.

**Justification For Why Not Lower Score:**

All reviewers agree to accept.

---

### Decision · Program_Chairs · 2024-01-16

Accept (poster)